# Robust Visual Compass Using Hybrid Features for Indoor Environments

**Ruibin Guo [1],\*, Keju Peng [1], Dongxiang Zhou [1] and Yunhui Liu [2]**

[1]  College of Electronic Science, National University of Defense Technology, Changsha 410073, China; keju009@nudt.edu.cn (K.P.); dxzhou@nudt.edu.cn (D.Z.)
[2]  Department of Mechanical and Automation Engineering, Chinese University of Hong Kong, Hong Kong 999077, China; yunhui.liu@gmail.com
\*  Correspondence: guoruibin64@126.com; Tel.: +86-731-845-03239

**Abstract:** Orientation estimation is a crucial part of robotics tasks such as motion control, autonomous navigation, and 3D mapping. In this paper, we propose a robust visual-based method to estimate robots' drift-free orientation with RGB-D cameras. First, we detect and track hybrid features (i.e., plane, line, and point) from color and depth images, which provides reliable constraints even in uncharacteristic environments with low texture or no consistent lines. Then, we construct a cost function based on these features and, by minimizing this function, we obtain the accurate rotation matrix of each captured frame with respect to its reference keyframe. Furthermore, we present a vanishing direction-estimation method to extract the Manhattan World (MW) axes; by aligning the current MW axes with the global MW axes, we refine the aforementioned rotation matrix of each keyframe and achieve drift-free orientation. Experiments on public RGB-D datasets demonstrate the robustness and accuracy of the proposed algorithm for orientation estimation. In addition, we have applied our proposed visual compass to pose estimation, and the evaluation on public sequences shows improved accuracy.

**Keywords:** visual compass; orientation estimation; hybrid features; plane tracking; vanishing direction; Manhattan World; RGB-D camera

---

## 1. Introduction

Robust orientation estimation is of great significance in robotics tasks such as motion control, autonomous navigation, and 3D mapping. Orientation can be obtained by utilizing carried sensors like the wheel encoder, inertial measurement unit (IMU) [1–4], or cameras [5–7]. Among these solutions, the visual-based method [8–11] is effective, as cameras can conveniently capture informative images to estimate orientation and position. In the past decades, many simultaneous localization and mapping (SLAM) systems [12,13] and visual odometry (VO) methods [14,15] have been proposed. Payá et al. [5] proposed a global description method based on Radon Transform to estimate robots' position and orientation with the equipped catadioptric vision sensor. These methods show good performance in estimating orientation from captured images. However, local and global maps' construction or loop detection is needed in these approaches to reduce drift error.

For most indoor environments, there exist many parallel and orthogonal lines and planes (called the Manhattan World (MW) [16]). These structural regularities are exploited in studies to estimate drift-free rotation without previous complex techniques (map reconstruction and loop closure) [17–19]. Since 3D geometric structures can easily be calculated by using the camera that provides both depth information and color image with 3 channels (red, green, and blue), called the RGB-D camera. The RGB-D camera has become a popular alternative to monocular and stereo cameras for the purpose

of rotation estimation, and estimation accuracy has been prominently improved by using the MW assumption with a RGB-D camera [20–23]. However, a major disadvantage of these MW-based methods is that the number of lines and planes used for tracking the MW axes must be no less than 2, which is the minimal sampling for 3 degrees of freedom (DoF). In practice, robots often encounter harsh environments without lines, and only one plane can be visible, resulting in failure in tracking MW axes to estimate the camera orientation. To address these issues, we select some frames as keyframes and exploit hybrid features (i.e., plane, line, and point) to compute the rotation matrix of each captured frame with respect to its reference keyframe instead of directly aligning with the global MW axes.

In this paper, we propose a robust and accurate approach for orientation estimation using RGB-D cameras. We detected and tracked the normal vectors of multiple planes from depth images, and we detected and matched the line and point features from color images. Then, by utilizing these hybrid features (i.e., plane, line, and point), we constructed a cost function to solve the rotation matrix of each captured frame. Meanwhile, we selected keyframes to reduce drift error and avoid directly aligning each frame with the global MW. Furthermore, we extracted the MW axes based on the normal vectors of orthogonal planes and the vanishing directions of parallel lines, and by aligning the current MW axes with the global MW axes, we refined the aforementioned rotation matrix of each keyframe and achieved drift-free orientation. Experiments showed that our proposed method produces lower drift error in a variety of indoor sequences compared to other state-of-the-art methods.

Our algorithm exploits hybrid features and adds a refinement step for keyframes, which can provide robust and accurate rotation estimation, even in harsh environments, as well as general indoor environments. The contributions of this work are as follows:

- We exploited the hybrid features (i.e., plane, line, and point), which provides reliable constraints in solving the rotation matrix for the majority of indoor environments.
- We refined keyframes' rotation matrix by aligning the current extracted MW axes with the global MW axes, which achieves drift-free orientation estimation.
- We evaluated our proposed approach on the ICL-NUIM and TUM RGB-D datasets, which showed robust and accurate performance.

## 2. Related Work

Pose estimation obtained by VO or V-SLAM systems has been extensively studied for the purpose of meaningful applications, such as autonomous robots and augmented reality. Rotation estimation is usually considered as a subproblem of pose estimation that consists of rotational and translation components. It has been gradually recognized by researchers that the main source of VO drift is inaccurate rotation estimation [19,20]. In the following discussion, we focus on rotation-estimation methods that exploit structural regularities with RGB-D cameras. These methods utilize surface normals, vanishing points (vanishing directions), or mixed constraints to compute camera orientation.

Surface-normal vectors were exploited to estimate camera orientation because their distribution on the unit sphere (or called Gaussian sphere) is regular and more likely around plane-normal vectors in the current environment, as shown in Figure 1. The work of Straub et al. [21] introduced the Manhattan-Frame model in the surface-normal space and proposed a real-time maximum a posteriori (MAP) inference algorithm to estimate drift-free orientation. Zhou et al. [23] developed a mean-shift paradigm to extract and track planar modes in surface-normal vector distribution on the unit sphere, and achieved drift-free behavior by registering the bundle of planar modes. In the work of Kim et al. [24], orthogonal planar structures were exploited and tracked with an efficient SO(3)-constrained mean-shift algorithm to estimate drift-free rotation. These surface-normal-based methods can provide stable and accurate rotation estimation if the number of observed orthogonal planes is not less than two.

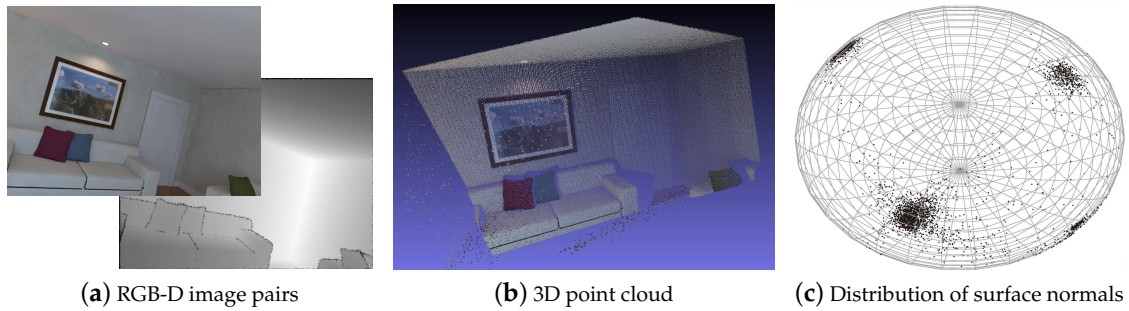

(**a**) RGB-D image pairs      (**b**) 3D point cloud      (**c**) Distribution of surface normals

**Figure 1.** Single RGB-D frame and distribution of surface normals corresponding to its 3D point cloud. (**a**) Frame 85 in 'LivingRoom0' sequence of ICL-NUIM dataset. (**b**) 3D point cloud obtained by back-projecting the depth information and coloring with aligned RGB pixels. (**c**) Distribution of surface normals on the unit sphere.

A vanishing point (VP) of a line is obtained by intersecting the image plane with a ray parallel to the world line and passing through the camera center, and it depends only on the direction of a line [25]. Two parallel lines determine a vanishing direction (VD), and the Euclidean 3D transformation of a VD is influenced only by rotation; the geometric relationships are shown in Figure 2, so VPs and VDs have been widely used for estimating rotation. Bazin et al. [17] proposed a three-line random-sample consensus (RANSAC) algorithm with the VP orthogonality constraint to estimate rotation. The work of Elloumi et al. [26] proposed a real-time pipeline for estimating camera orientation based on vanishing points for indoor navigation assistance on a smartphone. VP-based methods need a sufficient number of lines for estimating rotation, and accuracy performance is greatly affected by line-segment noise.

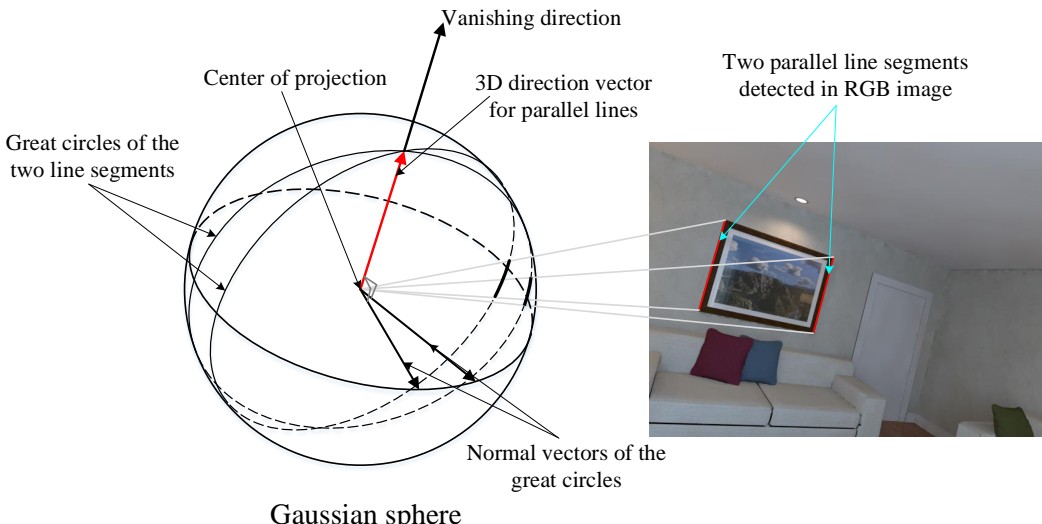

**Figure 2.** Three-dimensional geometric relationship between parallel lines and their vanishing direction. Gaussian sphere is a unit sphere on the center of a camera projection. Two parallel lines are projected onto the Gaussian sphere as two great circles, and vanishing direction is obtained by the cross project of these two great circles' normal vectors. Two parallel lines and their corresponding vanishing direction are drawn with red.

Hybrid approaches use both surface normals obtained in depth image and vanishing directions extracted in the RGB image to estimate rotation, which shows more robust performance. The method proposed by Kim et al. [22] exploited both line and plane primitives to deal with degenerate cases in surface-normal-based methods for stable and accurate zero-drift rotation estimation. In the work of Kim et al. [27], only a single line and a single plane in RANSAC were used to estimate camera orientation, and refinement is performed by minimizing the average orthogonal distance

from the endpoints of the lines parallel to the MW axes once the initial rotation estimation is found. Bazin et al. [17] introduced a related one-line RANSAC for situations where the horizon plane is known.

## 3. Proposed Method

We propose a robust visual compass that exploits hybrid features (i.e., plane, line, and point) to estimate camera orientation with the RGB and depth-image pairs. Our proposed method has two main steps: (1) rotation matrix is estimated by tracking the hybrid features for each frame with respect to the reference keyframe (tracking step); and (2) refine the keyframe's initial rotation matrix by aligning with the global MW axes to achieve drift-free orientation (refinement step). The overview of our proposed method is shown in Figure 3.

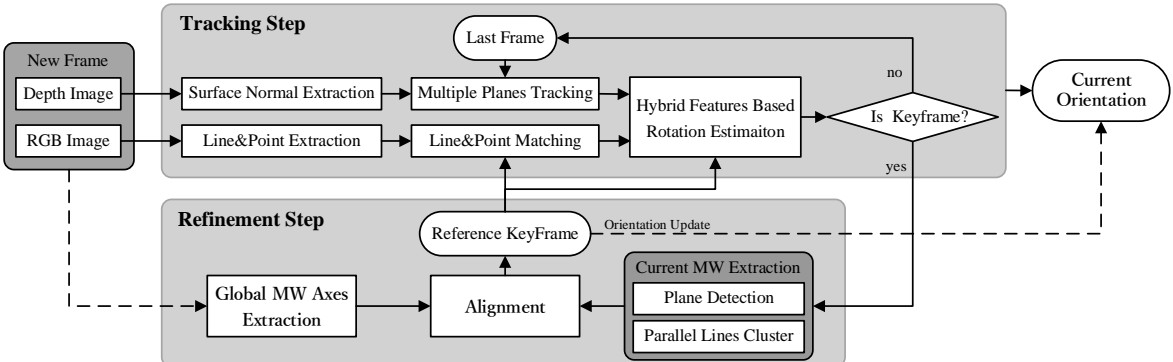

**Figure 3.** Overview of our visual compass. We estimate camera rotation by tracking the plane, line, and point features. We refine the keyframe orientation by aligning the current Manhattan World (MW) axes with the global MW axes. Global MW axes is extracted from the first captured RGB-D frame.

### 3.1. Rotation Estimation with Hybrid Features

We simultaneously tracked multiple planes, lines, and points in the current environment, and we utilized tracked hybrid features to construct a cost function for estimating the current rotation matrix relative to its reference keyframe. This can provide camera rotation even in uncharacteristic scenes where there are no rich texture or no consistent visible lines.

### 3.1.1. Multiple-Plane Detection and Tracking

We detected multiple planes from the depth image with a fast plane extraction algorithm [28]. The algorithm first constructs an initial graph that uniformly divides the depth image's point cloud into a set of nodes with size $H \times W$ in the image space. It then performs agglomerative hierarchical clustering on this graph to merge nodes belonging to the same plane. It final refines the extracted planes using pixel-wise region growing. With this approach, we obtain planes $P_i : (\mathbf{n}_i, d_i), i = 1, ..., m$, where $\mathbf{n}_i$ is the unit normal vector of the $i$-th plane, and $d_i$ is the distance to the origin of the camera co-ordinate system for the current frame.

We tracked the normal vector of each detected plane with a mean shift algorithm [23] that operates based on the density distribution of initial nodes' normals on the Gaussian sphere, as shown in Figure 4. We calculated the normal vector of each initial node by the least-square fitting method, and the depth image was preprocessed by a box filter to obtain the stable vectors. We used the previous frame's tracked (detected) normal vectors as an initial value, and then performed the mean shift algorithm in the tangent plane of the Gaussian sphere to obtain the tracked results. It should be noted that the parallel planes have the same plane normal vector; we class them as the same plane cluster for rotation estimation.

If we only have plane primitives to estimate rotation, rotation matrix $R$ with three degrees of freedom can be computed from no less than two such tracked norm vectors that are not parallel because each normal vector represents two independent constraints on $R$.

$$R = argmin \sum_{i=1,...,m} \left\| R \cdot \mathbf{n}_i^{ref} - \mathbf{n}_i^k \right\|_2^2 \tag{1}$$

where $\mathbf{n}_i^{ref}$ represents the $i$-th detected plane normal vector in the reference keyframe, and $\mathbf{n}_i^k$ represents the tracked result of the $i$-th plane in frame $k$.

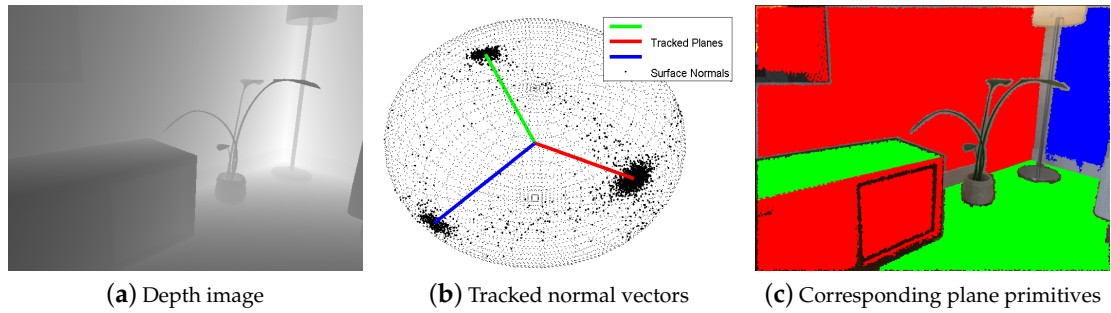

(**a**) Depth image     (**b**) Tracked normal vectors     (**c**) Corresponding plane primitives

**Figure 4.** Result of tracking planes: (**a**) Depth image of Frame 741 in the 'LivingRoom2' sequence of the ICL-NUIM dataset. (**b**) Tracked normal vectors from the Gaussian sphere; black dots represent the normals of initial nodes. (**c**) Plane primitives in current frame. Tacked normal vectors and their corresponding planes in image domain are represented with the same color.

### 3.1.2. Line and Point Detection and Matching

We used a linear-time Line Segment Detector (LSD) [29] to extract 2D line segments on the color image and obtain 2D line segments set $\mathbf{l} = \{l_i, i = 1, 2, ..., n\}$, where $l_i$ is the $i$-th line segment: $y = k_i x + b_i$, with starting point $\mathbf{u}_{si} = (u_{si}, v_{si})$ and ending point $\mathbf{u}_{ei} = (u_{ei}, v_{ei})$. Pixels belonging to the line segment $l_i$ are: $\tilde{\Re} = \{\mathbf{u} | \mathbf{u} \in l_i \wedge \mathbf{u} \in \Omega\}$, $\Omega$ is the image domain. Then, we reconstructed their corresponding 3D points set: $\mathbf{P}_l = \{\mathbf{p} | \mathbf{p} = \pi^{-1}(\mathbf{u}, d(\mathbf{u})), \mathbf{p} \in R^3 \wedge \mathbf{u} \in \tilde{\Re}\}$, where $d(\mathbf{u})$ represents the corresponding depth value of pixel $\mathbf{u}$ in the color domain, and $\pi^{-1}(\mathbf{u}, d(\mathbf{u})) = d(\mathbf{u})(\frac{u-c_x}{f_x}, \frac{v-c_y}{f_y}, 1)^T$ is the inverse projection function for a camera model, with $f_x, f_y$ being the focal lengths on the $x$ axis and $y$ axis, and $(c_x, c_y)^T$ is the camera's centre co-ordinates. Finally, the RANSAC method is used to fit 3D lines $L_i : (\mathbf{d}_i, \mathbf{p}_i)$, where $\mathbf{d}_i$ represents the $i$-th line's 3D direction, and $\mathbf{p}_i$ represents a point in this 3D line.

We match the lines that were respectively extracted from the current frame and the reference keyframe based on the Line Band Descriptor (LBD) [30], and two pairs of matching lines that are not parallel are needed to estimate the rotation matrix in case that there are only line primitives obtained in the current environment.

$$R = argmin \sum_{i=1...n} \left\| R \cdot \mathbf{d}_i^{ref} - \mathbf{d}_i^k \right\|_2^2 \tag{2}$$

where $\mathbf{d}_i^{ref}$ represents the $i$-th detected 3D line direction in the reference keyframe, and $\mathbf{d}_i^k$ represents the matching line direction in frame $k$.

In addition to the plane and line features, we extracted and matched the oriented fast and rotated brief (ORB) features for point tracking, as these features are extremely fast to compute and match, and they present good invariance to camera autogain, autoexposure, and illumination changes. We used

the epipolar constraint method to optimize the initial matching pairs by ORB descriptors; the optimized results can provide reliable constraints to estimate the camera pose.

$$\boldsymbol{R} = argmin \sum_{i...N} \left\| (\boldsymbol{R} \cdot \pi^{-1}(\mathbf{u}_i^{ref}, d(\mathbf{u}_i^{ref})) + \mathbf{t}) - \pi^{-1}(\mathbf{u}_i^k, d(\mathbf{u}_i^k)) \right\|_2^2 \tag{3}$$

where $\mathbf{u}_i^{ref}$ represents the 2D position of *i*-th detected ORB point feature in the reference keyframe and $\mathbf{u}_i^k$ represents the 2D position of the matching point in frame *k*. It should be noted that translation component $\mathbf{t}$ could be obtained by solving Equation (3), but we did not consider it, as our visual compass mainly focused on camera orientation.

### 3.1.3. Robust Rotation Estimation

We jointly utilized the tracked planes, lines, and points in the current environment to estimate the rotation matrix with respect to the reference keyframe. Rotation matrix $\boldsymbol{R}$ can be computed by solving:

$$\boldsymbol{R} = argmin\left( \sum_{i=1...m} \lambda_i^P \left\| \boldsymbol{R} \cdot \mathbf{n}_i^{ref} - \mathbf{n}_i^k \right\|_2^2 + \sum_{i=1...n} \lambda_i^L \left\| \boldsymbol{R} \cdot \mathbf{d}_i^{ref} - \mathbf{d}_i^k \right\|_2^2 + \sum_{i=1...N} \left\| (\boldsymbol{R} \cdot \boldsymbol{P}_i^{ref} + \mathbf{t}) - \boldsymbol{P}_i^k \right\|_2^2 \right) \tag{4}$$

where $\lambda_i^P$ represents the number of pixels contained in the *i*-th tracked plane, and $\lambda_i^L$ represents the number of pixels contained in the *i*-th tracked line, 3D points $\boldsymbol{P}_i^{ref} = \pi^{-1}(\mathbf{u}_i^{ref}, d(\mathbf{u}_i^{ref}))$, and $\boldsymbol{P}_i^k = \pi^{-1}(\mathbf{u}_i^k, d(\mathbf{u}_i^k))$.

Cost function Equation (4) contains three parts, corresponding to plane, line, and point constraints. A stable and accurate rotation matrix can be solved by minimizing Equation (4) with line and plane constraints jointly in the texture-less environment that few points tracked, and the point constraints ensure that the rotation estimation is reliable in the scenes that no consistent lines or only one plane to be visible.

**Keyframe Selection:** By the tracking step, we constantly know the number of the tracked planes, lines and points for each frame. If there is only one tracked plane with the condition that the number of normal vectors on the Gaussian sphere around this tracked normal vector is too low, and the number of tracked points is less than a threshold, we reuse the fast plane extraction method and Line Segment Detector (LSD) method to detect planes and lines in the current frame. If the number of redetected orthogonal planes and lines is larger than 2, this frame is selected as a keyframe and performs the following refinement step.

### 3.2. Drift-Free Orientation Estimation

The previous tracking step estimates the rotation matrix between the current frame and its reference keyframe, and it is obvious that the accuracy of the reference keyframe's orientation directly affects the accuracy of the current frame's rotation matrix. To reduce drift error, we sought global MW axes in the first frame and refined each keyframe's orientation by aligning the current extracted MW axes with the global MW axes to achieve drift-free rotation in MW scenes. We sought current and global MW axes based on the plane normal vectors, and the vanishing directions of the parallel lines. Plane normal vectors can be directly obtained by the previous fast plane extraction method, and we propose a novel vanishing direction extraction method as follows.

### 3.2.1. Vanishing Direction Extraction

To extract accurate VDs, we need to cluster lines that are parallel in the real world. We used the simplified Expectation–Maximization (EM) clustering method to group image lines and compute their corresponding 3D direction vectors. The original EM algorithm iterates the expectation and the maximization steps. In our simplified algorithm, we skip the expectation phase and roughly cluster the lines based on the K-means method [31], with the Euclidean distance of all extracted lines' 3D

directions that are represented as 3D points. In the maximization phase, direction vectors are estimated by maximizing the objective function:

$$\mathbf{d}_k^v = argmax \prod_i p(\mathbf{d}_k^v | \mathbf{l}_i^{(k)}) \tag{5}$$

where $\mathbf{l}_i^{(k)}$ represents the *i*-th 2D line segment in the *k*-th initial classification, $\mathbf{d}_k^v$ represents the VD of the *k*-th initial classification to be optimized, and the $p(\mathbf{d}_k^v | \mathbf{l}_i^{(k)})$ represents the posterior likelihood of the VD.

Using the Bayes formula, the posterior likelihood of the VD is expressed as:

$$p(\mathbf{d}_k^v | \mathbf{l}_i^{(k)}) = \frac{p(\mathbf{l}_i^{(k)} | \mathbf{d}_k^v) p(\mathbf{d}_k^v)}{p(\mathbf{l}_i^{(k)})} \tag{6}$$

where $p(\mathbf{l}_i^{(k)} | \mathbf{d}_k^v)$ represents the prior probability of the VD, and $p(\mathbf{d}_k^v)$ models the potential knowledge of the VD before we obtain the measurement. If we know nothing, $p(\mathbf{d}_k^v)$ is defined as uniform distribution with a constant value. Therefore, the VD can be obtained by maximizing prior probability:

$$\mathbf{d}_k^v = argmax \prod_i p(\mathbf{d}_k^v | \mathbf{l}_i^{(k)}) = argmax \sum_i \log p(\mathbf{l}_i^{(k)} | \mathbf{d}_k^v) \tag{7}$$

Prior probability $p(\mathbf{l}_i^{(k)} | \mathbf{d}_k^v)$ is defined as:

$$p(\mathbf{l}_i^{(k)} | \mathbf{d}_k^v) = \frac{1}{\sqrt{2\pi\sigma_k^2}} \left( \frac{-(\mathbf{l}_i^{(k)\mathbf{T}} \mathbf{K} \mathbf{d}_k^v)^2}{2\sigma_k^2} \right) \tag{8}$$

where **K** represents the internal camera parameters. Equation (8) reflects the fact that the vanishing direction is perpendicular to the plane normal of a great circle that is determined by an image line $\mathbf{l}_i^{(k)}$ and the center of the projection of the camera, as shown in Figure 2.

Maximizing objective function Equation (7) is equivalent to solving a weighted least-squares problem for each $\mathbf{d}_k^v$:

$$\mathbf{d}_k^v = argmin \sum_i \frac{length(\mathbf{l}_i^{(k)})}{max(length(\mathbf{l}^{(k)}))} \cdot (\mathbf{l}_i^{(k)\mathbf{T}} \mathbf{K} \mathbf{d}_k^v)^2 \tag{9}$$

where $length(\mathbf{l}_i^{(k)})$ represents the length of the *i*-th line, and $max(length(\mathbf{l}^{(k)}))$ represents the maximum line length in rough cluster *k*. Length coefficient is considered in the term because the longer the lines are, the more reliable they are. By solving Equation (9), we can obtain the initial vanishing direction and the residual for each line. To obtain a more accurate vanishing direction, we discarded lines with a larger residual than a threshold and added additional optimization. Optimized parallel lines are used to estimate the final vanishing directions. Figure 5 shows two results of parallel-line clustering obtained by our proposed method.

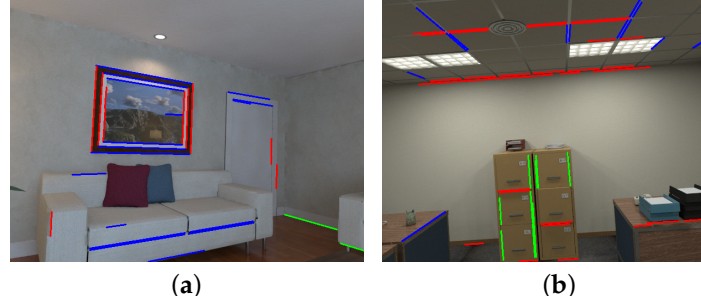

(**a**)                      (**b**)

**Figure 5.** Results of parallel-line clustering that are used to compute vanishing directions: (**a**) the 120-th image in 'Living Room 2' sequence. (**b**) the 64-th image in 'Office Room 1' sequence. Lines with the same color are parallel in the real world. The vanishing directions obtained by these parallel lines can provide accurate and reliable constraints for MW extraction.

### 3.2.2. Global Manhattan World Seeking

The Manhattan World assumption is suitable in human-made indoor environments, and we sought global MW axes based on plane normal vectors and the vanishing directions from the first frame. MW axes can be expressed as columns of a 3D rotation matrix $R = [\mathbf{r}_1^g \ \mathbf{r}_2^g \ \mathbf{r}_3^g] \in SO(3)$.

We first set the the detected plane normal vectors and the vanishing directions from the parallel lines as the candidate MW axes, which return a redundant set. In fact, most of the pixels in the frame typically belong to the planes and lines that determine the dominant MW axes, and we sought the plane that contained the largest pixels, and set its plane normal vector $\mathbf{r}_1$ as the first MW axis. The remaining two axes $\mathbf{r}_2$ and $\mathbf{r}_3$ are determined based on the orthogonality constraint with the first axis and the number of the pixels belonging to the detected planes or the parallel lines.

If we detect three mutually orthogonal planes in the first frame, we directly set their plane-normal vectors as the MW axes. In the case that there are only two orthogonal planes, the third axis is determined by the vanishing direction from the parallel lines that is orthogonal with the two previous plane-normal vectors. Similarly, if only one plane is detected in the first frame, we sought the remaining two MW axes by the vanishing directions from the orthogonal parallel lines.

Initial MW axes $[\mathbf{r}_1 \ \mathbf{r}_2 \ \mathbf{r}_3]$, obtained by the previous step, are not strictly orthogonal. We maintained orthogonality by reprojecting the MW axes onto the closest matrix on $SO(3)$. Each axis is reweighted by a factor $\lambda_i$ that is determined by the number of pixels belong to this axis' corresponding planes or parallel lines. The final global MW axes are obtained by using singular=value decomposition (SVD):

$$\begin{bmatrix} \mathbf{r}_1^g & \mathbf{r}_2^g & \mathbf{r}_3^g \end{bmatrix} = \mathbf{U}\mathbf{V}^T \tag{10}$$

where $[\mathbf{U}, \mathbf{D}, \mathbf{V}] = SVD([\lambda_1 \mathbf{r}_1 \ \lambda_2 \mathbf{r}_2 \ \lambda_3 \mathbf{r}_3])$ and factor $\lambda_i$ describes how certain the observation of a direction is.

### 3.2.3. Keyframe Orientation Refinement

For each keyframe, we refined its rotation matrix by aligning the current extracted MW axes with the global MW axes. We first used the fast plane-extraction algorithm and our proposed spatial-line direction estimation method to extract plane-normal vectors and vanishing directions in the current keyframe. We then extracted the current MW axes ($\mathbf{r}_i^c, i = 1, 2, 3$) by using the same method that we used to extract global MW axes. We finally determine the corresponding pairs based on the Euclidean distance between vector $\mathbf{r}_i^c$ and $\mathbf{R}\mathbf{r}_j^g$, where $\mathbf{R}$ represents the rotation matrix obtained by tracking steps for the current keyframe. Refined rotation matrix $\mathbf{R}^r$ is computed by solving:

$$\mathbf{R}^r = argmin \sum_{i=1,2,3} \left\| \mathbf{R}^r \mathbf{r}_i^g - \mathbf{r}_i^c \right\|_2^2 \tag{11}$$

where $\mathbf{r}_i^g$ and $\mathbf{r}_i^c$ represent the *i*-th global MW axis and the current extracted corresponding MW axis.

## 4. Results

We evaluate our proposed approach on the synthetic dataset (ICL-NUIM [32]), real-world dataset (TUM RGB-D [33]), and pose-estimation application, respectively. All experiments were run on a desktop computer with an Intel Core i7, 16 GB memory, and Ubuntu 16.04 platform.

- The *ICL-NUIM* dataset is a collection of handheld RGB-D camera sequences within synthetically generated environments. These sequences were captured in a living room and an office room with perfect ground-truth poses to fully quantify the accuracy of a given visual odometry or SLAM system. Depth and RGB noise models were used to alter the ground images to simulate realistic sensor noise. There are sequences that are captured in the environment with low texture and only one visible plane, which makes it hard to estimate rotation for whole images in this sequence.
- The *TUM RGB-D* dataset is a famous benchmark to evaluate the accuracy of a given visual odometry or visual SLAM system. It contains various indoor sequences captured from the Kinect RGB-D sensor. The sequences were recorded in real environments at a frame rate of 30 Hz with a $640 \times 480$ resolution, and their ground-truth trajectories were obtained from a high-accuracy motion-capture system. The TUM dataset is more challenging than the ICL dataset because it has some blurred images and inaccurate alignment image pairs that make it difficult to estimate the rotation matrix.

We compared our proposed approach with two state-of-the-art MW-based methods proposed by Zhou et al. [23] and Kim et al. [27], namely, orthogonal planes based rotation estimation (OPRE) and 1P1L. OPRE estimates absolute and drift-free rotation by exploiting orthogonal planes from depth images. 1P1L estimates 3DoF drift-free rotational motion with only a single line and plane in the Manhattan world. We used the average value of the absolute rotation error (ARE) in degrees as the performance metric for the entire sequences:

$$\mathbf{ARE}.average = \frac{1}{N} \sum_{i=1}^{N} \arccos\left(\frac{tr(\mathbf{R}_i^T \cdot \mathbf{R}_i^g) - 1}{2}\right) \times 57.3 \tag{12}$$

where $tr()$ denotes the trace of a matrix, $\mathbf{R}_i$ and $\mathbf{R}_i^g$ represent the estimated and ground rotation matrix for the *i*-th frame, respectively, and $N$ represents the number of frames in the tested sequence.

### 4.1. Evaluation on Synthetic Dataset

We first tested the performance of our proposed algorithm on the ICL-NUIM dataset, and measured the average ARE in degrees for each sequence; evaluation results are shown in Table 1. The smallest average ARE values are bolded, which reveals that our proposed method is more accurate than the two other methods. For example, in 'Office Room 0', the average ARE of our proposed method is 0.16 degrees, while that of 1P1L and OPRE is 0.37 and 0.18 degrees, respectively. Our method outperformed the others in all cases in the ICL-NUIM benchmark. The main reason is that we jointly exploited the plane, line, and point features to estimate camera orientation even when the camera moves in scenes with no consistent lines, or where only one plane is visible; this is illustrated in Figure 6. In 'Living Room 0', the OPRE method failed to estimate the rotation for the entire sequence because only one plane can be visible in some frames; we marked the result as '×' in Table 1. The last column in Table 1 shows the number of frames in the current sequence.

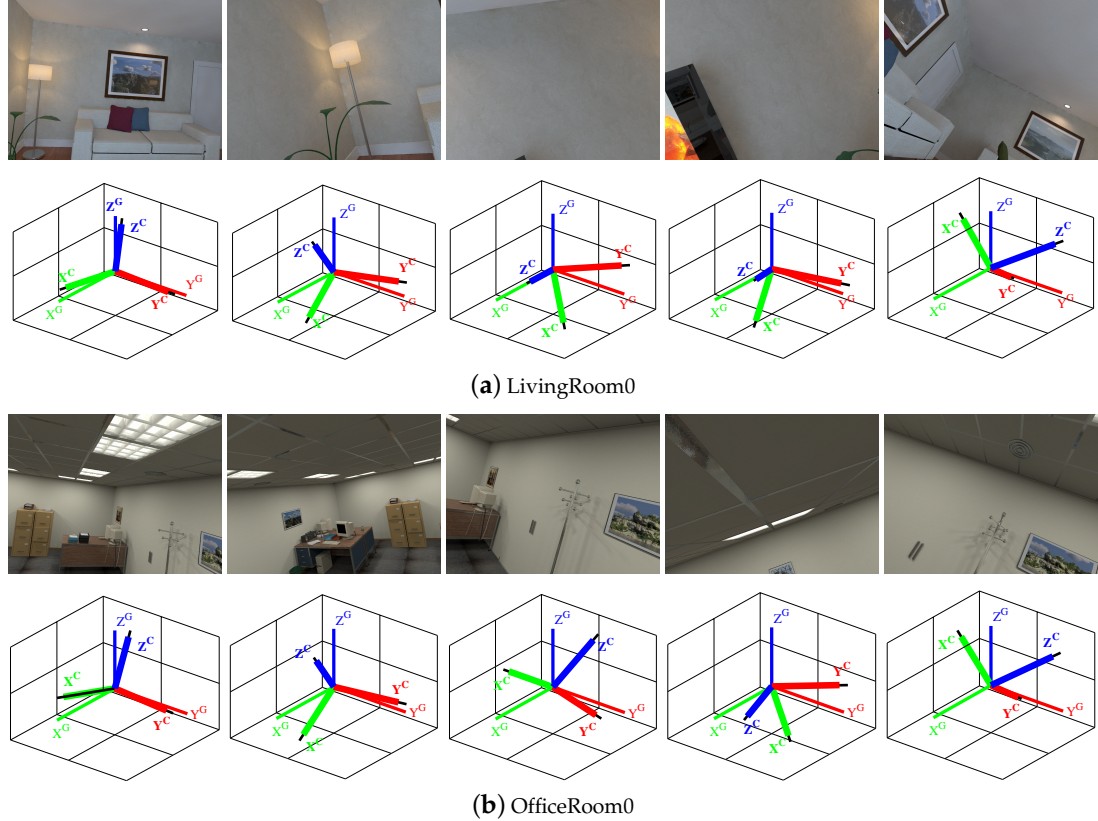

**(a)** LivingRoom0

**(b)** OfficeRoom0

**Figure 6.** Results of camera orientation estimated by our proposed method on the ICL-NUIM dataset: (**a**) 'Living Room 0' sequence. (**b**) 'Office Room 0' sequence. The estimated orientation of each frame is shown in the bottom of the RGB image. Colored thick and thin lines respectively denote current orientation and global MW axes (determined in the first frame); black lines represent ground orientation.

**Table 1.** Comparison of average absolute rotation error (ARE, degrees) on the ICL-NUIM dataset.

| Sequence | Proposed | No Refinement | 1P1L | OPRE | Frames |
|---|---|---|---|---|---|
| Living Room 0 | **0.22** | 0.53 | 0.31 | $\times$ | 1508 |
| Living Room 1 | **0.25** | 0.55 | 0.38 | 0.97 | 965 |
| Living Room 2 | **0.23** | 1.26 | 0.34 | 0.49 | 880 |
| Living Room 3 | **0.35** | 1.69 | 0.35 | 1.34 | 1240 |
| Office Room 0 | **0.16** | 1.32 | 0.37 | 0.18 | 1507 |
| Office Room 1 | **0.17** | 0.44 | 0.37 | 0.32 | 965 |
| Office Room 2 | **0.26** | 1.29 | 0.38 | 0.33 | 880 |
| Office Room 3 | **0.14** | 0.43 | 0.38 | 0.21 | 1240 |

The MW assumption is sufficiently suitable for the ICL-NUIM benchmark and we used the refinement step to achieve a drift-free rotation matrix. To clearly show the effect of the refinement step, we measured the ARE values in degrees for all sequences by our method without a refinement step, which corresponds to the 'No Refinement' column in Table 1. We recorded the values of absolute rotation error (ARE) for each frame in the 'Living Room 0' sequence, and the final rotation drift with and without refinement was 0.34 and 1.43 degrees, respectively, as shown in Figure 7. This demonstrates that the refinement step can effectively reduce rotation drift.

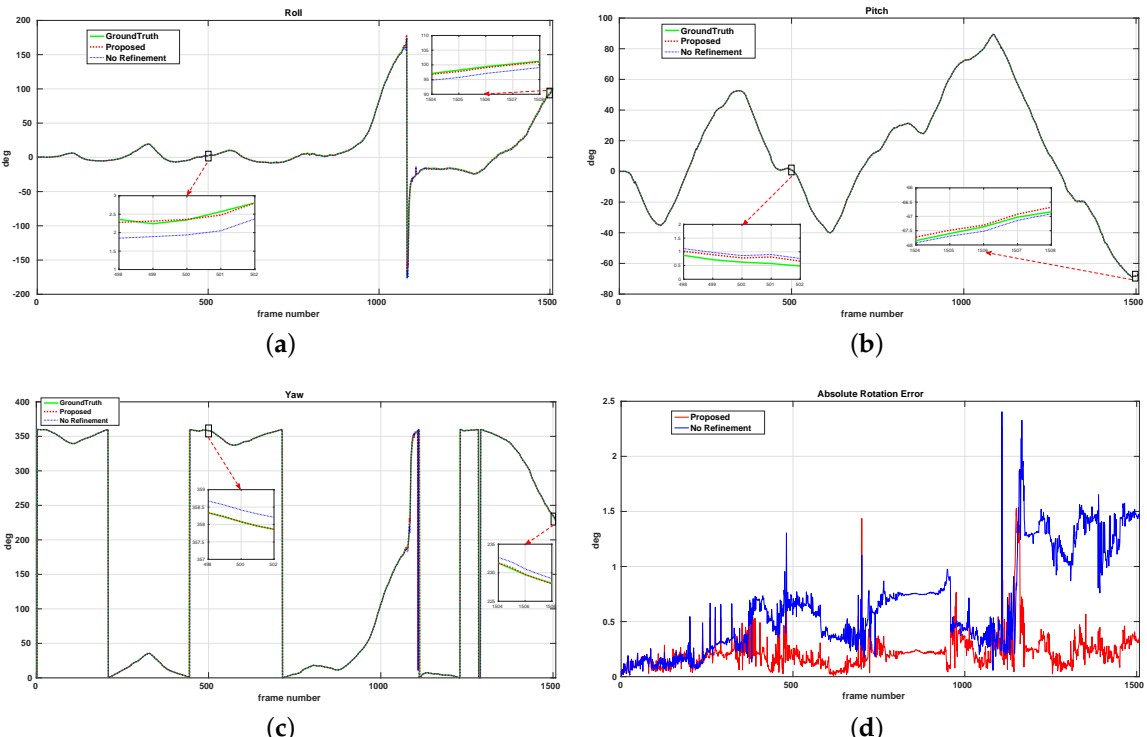

**Figure 7.** Performance evaluation for refinement on the 'Living Room 0' sequence: (**a**–**c**) Roll, pitch, and yaw angles estimated by the proposed method with and without a refinement step for each frame. (**d**) Absolute rotation errors for our proposed methods with and without a refinement step. This shows that the refinement step can effectively reduce rotation drift.

### 4.2. Evaluation on Real-World Data

We compared the performance of our proposed algorithm with other two methods on seven real-world TUM RGB-D sequences that contained structural regularities. Comparison results are shown in Table 2. We provide the average ARE for rotation estimation, and the smallest values are indicated in bold. Our proposed method showed better performance in low-texture environments such as 'fr3_struc_notex' and 'fr3_cabinet' because we used hybrid features to estimate orientation, as shown in Figure 8. Our method can also provide a more accurate rotation matrix in an environment with imperfect MW structure like 'fr3_nostruc_tex' and 'fr3_nostruc_notex', whereas OPRE fails because it requires at least two orthogonal planes to estimate camera orientation.

**Table 2.** Comparison of average ARE (degrees) on TUM RGBD Dataset.

| Sequence | Proposed | No Refinement | 1P1L | OPRE | Frames |
|----------|----------|---------------|------|------|--------|
| fr3_nostruc_notex | **1.22** | 1.22 | 1.51 | $\times$ | 90 |
| fr3_nostruc_tex | **1.89** | 1.89 | 2.15 | $\times$ | 448 |
| fr3_struc_notex | **1.20** | 1.62 | 1.96 | 3.01 | 965 |
| fr3_struc_tex | **0.74** | 1.03 | 2.92 | 3.81 | 905 |
| fr3_cabinet | **1.48** | 2.78 | 2.48 | 2.42 | 926 |
| fr3_large_cabinet | **1.87** | 3.95 | 2.04 | 36.34 | 980 |
| fr3_long_office | **1.51** | 3.58 | 1.75 | 4.99 | 2486 |

The result of refinement performance on the 'fr3_cabinet' sequence is shown in Figure 9. Final rotation drift with and without refinement was 1.30 and 1.62 degrees, respectively. It is clear that our refinement step can effectively reduce drift error. The average ARE values computed by our proposed algorithm with and without refinement step were the same in sequences 'fr3_nostruc_tex'

and 'fr3_nostruc_notex'. The reason is that there were no perfect global MW axes extracted in the first frame, and the refinement step was not implemented.

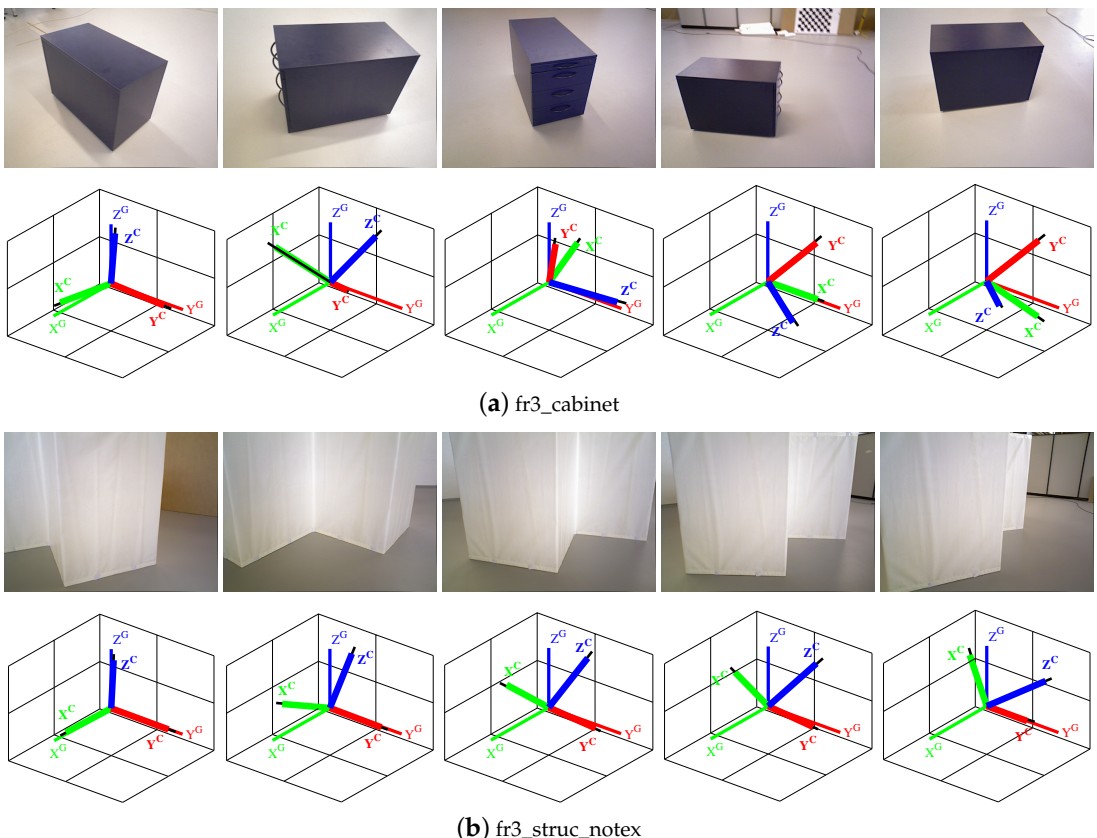

(**a**) fr3_cabinet

(**b**) fr3_struc_notex

**Figure 8.** Results of camera orientation estimated by our proposed method on the TUM RGB-D dataset: (**a**) 'fr3_cabinet' sequence. (**b**) 'fr3_struc_notex' sequence. Estimated orientation of each frame is shown in the bottom of the RGB image. Colored thick and thin lines respectively denote current orientation and global MW axes (determined in the first frame); black lines represent ground orientation.

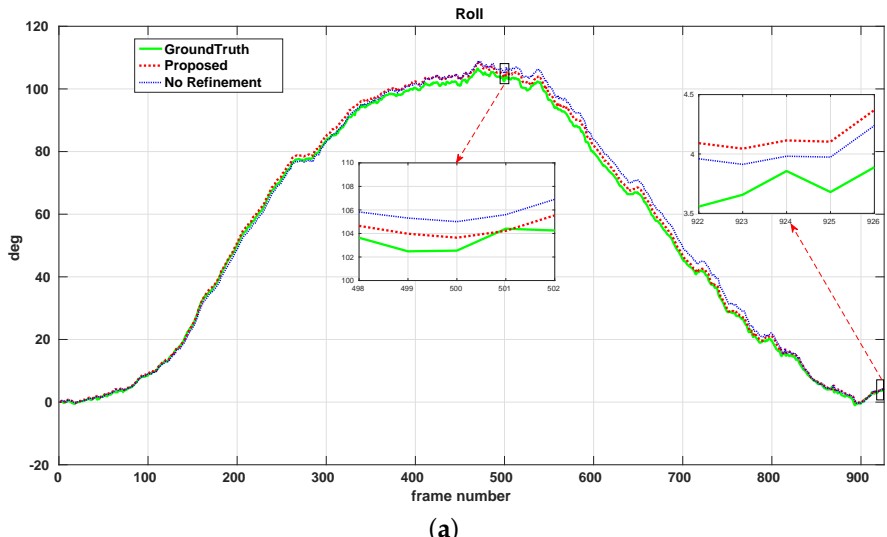

(**a**)

**Figure 9.** *Cont.*

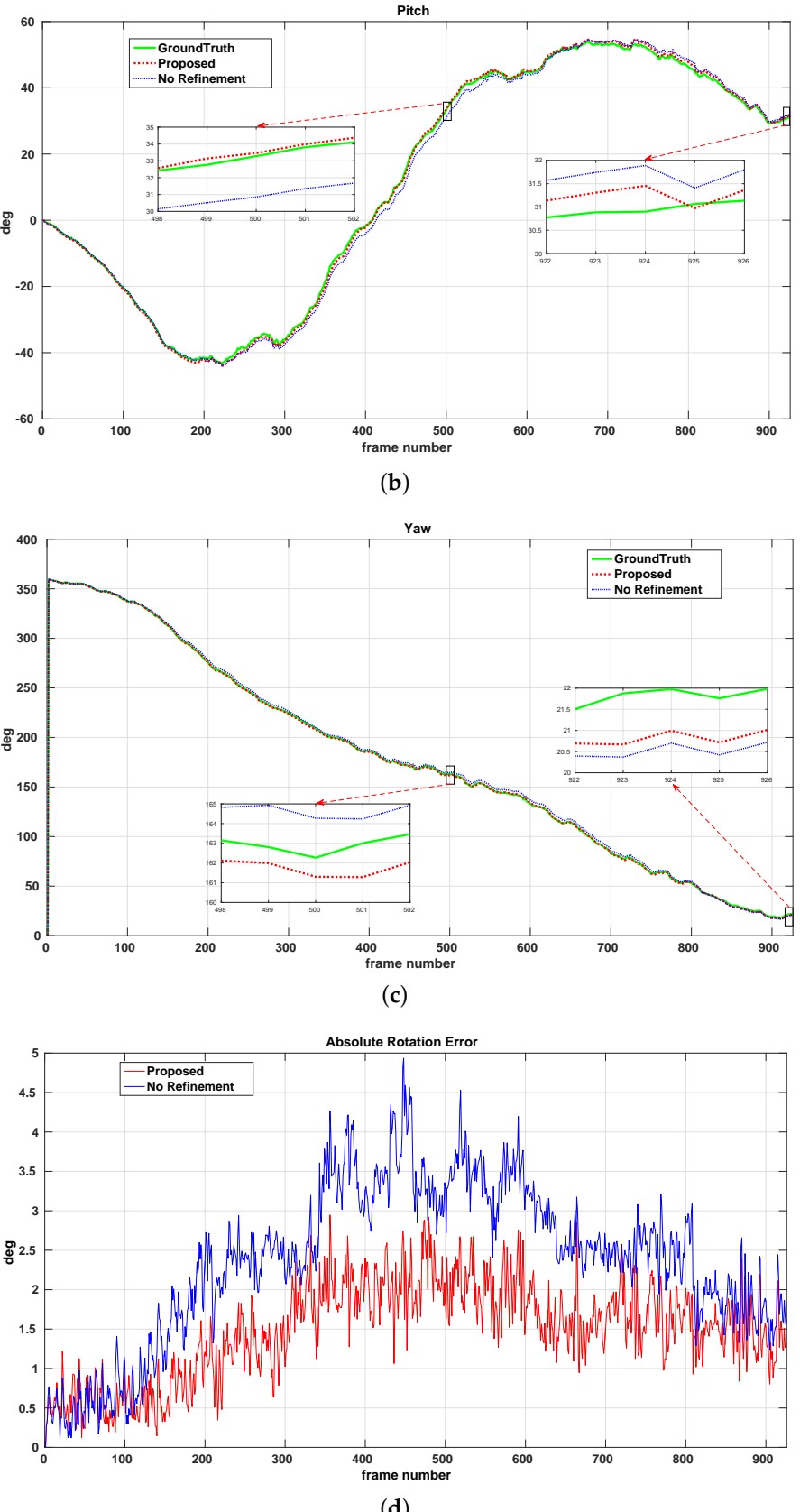

**Figure 9.** Performance evaluation for refinement on the 'fr3_cabinet' sequence: (**a**–**c**) Roll, pitch, and yaw angles estimated by our proposed method with and without a refinement step for each frame. (**d**) Absolute rotation errors for the proposed methods with and with refinement step. This shows that the refinement step can effectively reduce rotation drift.

### 4.3. Application to Pose Estimation

To further verify the practicability of our proposed visual compass, we used it for pose-estimation application and recorded the estimated trajectories. The three-dimensional pose has six degrees of freedom (DoF) and it consists of 3-DoF rotation and 3-DoF translation. As our proposed visual compass method can provide accurate rotation estimation, the key to performing pose localization is to estimate the translation component. We first detected and tracked ORB feature points to obtain point correspondences. Then, we recovered the 3-DoF translational motion of the images by minimizing:

$$t = argmin(\sum_{i=1...N} \left\| (R^{fixed} \cdot P_i^{ref} + \mathbf{t}) - P_i^k \right\|_2^2) \tag{13}$$

where $R^{fixed}$ represents the rotation matrix between the reference image and the current image, and it is obtained by our proposed visual compass, Three-dimensional points $P_i^{ref}$ and $P_i^k$ are described in Equation (4).

We tested pose estimation on four datasets, "Living Room 2", "Office Room 3", "fr3_struc_tex", and "fr3_nostruc_tex". These datasets provide the ground-truth pose for each image; we measured the root mean squared error (RMSE) of the absolute translational error (ATE) and compared it with state-of-the-art approaches, namely, ORB_SLAM [12], dense visual odometry (DVO) [15], and line-plane based visual odometry (LPVO) [22]. The comparison of ATE.RMSE is shown in Table 3; the smallest error for each sequence is indicated in bold. Estimated trajectories are shown in Figure 10.

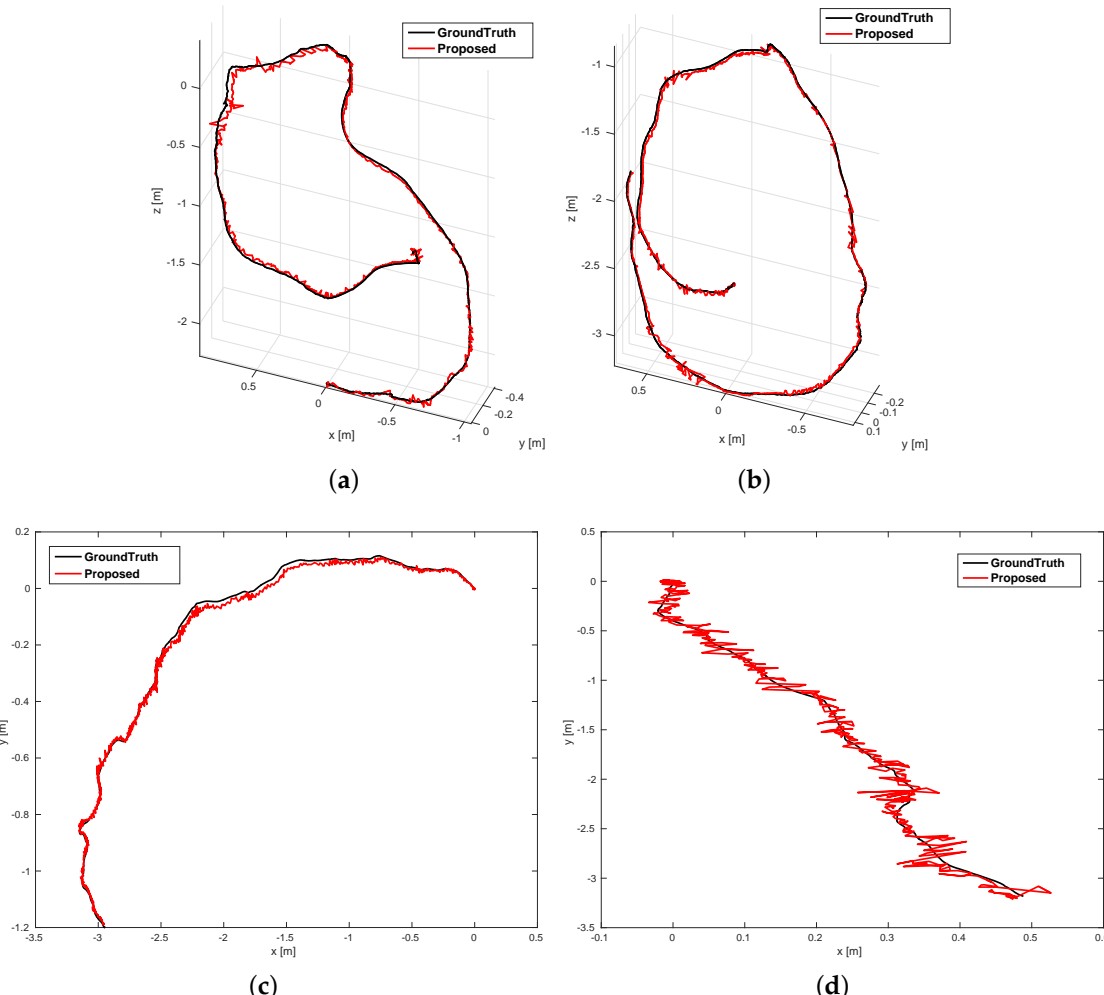

**Figure 10.** Estimated trajectories with the proposed (red) and ground truth (black) on four sequences: (**a**) Living Room2, (**b**) Office Room 3, (**c**) fr3_struc_tex, (**d**) fr3_nostruc_tex.

**Table 3.** Comparison of ATE.RMSE (unit: m).

| Sequence | Proposed | ORB-SLAM2 | DVO | LPVO |
|----------|----------|-----------|-----|------|
| Living Room 2 | **0.025** | 0.028 | 0.375 | 0.034 |
| Office Room 3 | **0.021** | 0.065 | 0.079 | 0.030 |
| fr3_struc_tex | **0.017** | 0.024 | 0.048 | 0.174 |
| fr3_nostruc_tex | **0.046** | 0.052 | 0.073 | × |

## 5. Conclusions

We proposed a visual-based method to estimate robot orientation with RGB-D cameras. We exploited hybrid features providing reliable constraints to construct cost function for solving the initial rotation matrix. We presented a vanishing direction extraction method based on parallel lines and combined it with detected plane normals to seek global and current Manhattan World axes. We refined the orientation matrix of the selected keyframe with respect to the global MW axes, and achieved drift-free orientation. The proposed algorithm was tested on both synthetic as well as real-world publicly available RGB-D datasets, and we compared it with two state-of-the-art methods for orientation estimation. The results demonstrated the accuracy and robustness of our proposed method. Furthermore, we applied the proposed algorithm to pose estimation and recovered the translational motion by giving absolute camera orientation; evaluation on the public sequences showed improved accuracy. In summary, the proposed algorithm showed good performance in Manhattan World scenes, and it has significant applications on mobile robotics. In the future, we will exploit hybrid features to perform pose location and 3D mapping that can provide maps with a geometric structure and more robust pose estimation in less-textured and -structured environments, and we will try to optimize the refinement step for not only pure Manhattan Worlds but also more general environments like Mixtures of Manhattan Worlds [34].

**Author Contributions:** Methodology, R.G.; resources, R.G.; software, R.G.; supervision, K.P. and D.Z.; validation, K.P. and Y.L.; writing—original draft, R.G.; writing—review and editing, D.Z. and Y.L.

**Funding:** This research received no external funding.

**Conflicts of Interest:** The authors declare no conflict of interest.

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
