# Peer review of "Robust Visual Compass Using Hybrid Features for Indoor Environments"

_electronics, doi:10.3390/electronics8020220_

Round 1
Reviewer 1 Report
Authors present an approach to image orientation based on RGB-D images sustained by a minimization algorithm in terms of visual object detection and tracking.
Despite the fact that the manuscript is sound and well written, I personally believe that the study needs more real data results apart from merely testing orientation estimation within public datasets. I strongly encourage authors to present some extended experiments, for instance, to perform pose localization with this sort of orientation measurement. The consulted datasets should also provide ground thruth to test image pose localization estimated from the orientation compass presented here.
Apart from this, some other minor comments:
l14. Please start numbering sections from 1.
l15. The first statement should be also sustained by up-to-date cites for the metioned fields. There is an endless list of works published in the MDPI database, within Journals such as Sensors or Remote Sensing. Consider also appearance-based methods for orientation retrieval, in order to increase the discussion:
"Estimating the position and orientation of a mobile robot with
respect to a trajectory using omnidirectional imaging and global
appearance". L. Payá et al. PLOS ONE.
l18. Same comment applies for an example of visual methods to estimate rotation and orientation.
Fig7(a). Resolution and/or size is too low to perceive the little zooms. Please increase size and/or resolution of this image, or even show separately roll, pitch and yaw.
Fig.9(a). Same comment.
Author Response
Dear Reviewer,
We quite appreciate your consideration and insightful comments concerning our manuscript entitled "Robust Visual Compass Using Hybrid Features for Indoor Environments". These comments are very valuable and helpful for improving the quality of our manuscript. We have studied the comments carefully and have revised the manuscript exactly according to these comments. We hope these revisions can meet with approval. The main revisions corresponding to your comments are descripted in attachment: "cover letter for reviewer 1.pdf".
Best Regards!
Yours sincerely,
Ruibin Guo

Reviewer 2 Report
Major comment: The authors have to include more real world examples.
The method which is presented relies on methods with a certain measure of uncertainty, questions occur with respect to including additional sources of inaccuracy (for example, plane extraction algorithm (is its accuracy evaluated), when does this approach fails to compute proper plane representatives of arbitrary scenes, how robust is this approach against noise, etc...
In the introduction the authors state common methods like ORB_SLAM, LSD_SLAM, SVO and DVO in order to solve orientation estimation problems in robotic applications and mention, that these methods are computational expensive. - How does the proposed approach compares to such methods in terms of accuracy and computational efficiency?
Would it be possible for the authors to show limitations of the approach?
Figures are of low resolution, could be improved.
Author Response
Dear Reviewer,
We quite appreciate your consideration and insightful comments concerning our manuscript entitled "Robust Visual Compass Using Hybrid Features for Indoor Environments". These comments are very valuable and helpful for improving the quality of our manuscript. We have studied the comments carefully and have revised the manuscript exactly according to these comments. We hope these revisions can meet with approval. The main revisions corresponding to your comments are descripted in attachment: "cover letter for reviewer 2.pdf".
Best Regards!
Yours sincerely,
Ruibin Guo

Round 2
Reviewer 1 Report
Authors have accomplished the suggestions sufficiently. However, some slight details have to be tackled yet:
- A more specific mention to the kind of experimental set has to be given in the Abstract. Please state clear that you performed orientation and pose estimation experiments at the end of the Abstract.
- Similarly, in the Related Work, now you include pose estimation, a couple of reference such as the two following, should be added and commented.
https://www.mdpi.com/2072-4292/11/1/73
https://www.mdpi.com/1424-8220/17/2/325/htm
- Finally, the Conclusions should be extended a bit by giving some more detail of what the work has done. They are very short now. Readers should get a quick idea of what this manuscript has accomplished. Please just extend in a few lines more, some details about the major contributions and benefits of the approach, and comment on the experimental setup and the results extracted.
Author Response
Dear Reviewer,
We quite appreciate your consideration and insightful comments concerning our manuscript entitled "Robust Visual Compass Using Hybrid Features for Indoor Environments". These comments are very valuable and helpful for improving the quality of our manuscript. We have studied the comments carefully and have revised the manuscript exactly according to these comments. We hope these revisions can meet with approval. The main revisions corresponding to your comments are detailed in attachment "cover letter for reviewer1_Round2.pdf".
Thank you again for your help. If there are other problems, please don't hesitate to contact me!
Best Regards!
Yours sincerely,
Ruibin Guo

Reviewer 2 Report
The authors did improved the document with respect to the comments provided.
Author Response
Dear Reviewer,
We quite appreciate your consideration and insightful comments concerning our manuscript entitled "Robust Visual Compass Using Hybrid Features for Indoor Environments". Thank you again for your help.
Best Regards!
Yours sincerely,
Ruibin Guo